# The Cytotoxicity and Clearance of Mutant Huntingtin and Other Misfolded Proteins

**DOI:** 10.3390/cells10112835

**Published:** 2021-10-21

**Authors:** Austin Folger, Yanchang Wang

**Affiliations:** Department of Biomedical Sciences, College of Medicine, Florida State University, Tallahassee, FL 32306, USA; austin.folger@med.fsu.edu

**Keywords:** protein misfolding, mutated Huntingtin, ubiquitin-proteasome system (UPS), Cdc48, autophagy

## Abstract

Protein misfolding and aggregation are implicated in many neurodegenerative diseases. One of these diseases is Huntington’s, which is caused by increased glutamine-encoding trinucleotide repeats within the Huntingtin gene. Like other misfolded proteins, mutated Huntingtin proteins with polyglutamine expansions are prone to aggregation. Misfolded proteins exist as soluble monomers, small aggregates, or as large insoluble inclusion bodies. Misfolded protein aggregates are believed to be cytotoxic by stressing the protein degradation machinery, disrupting membrane structure, or sequestering other proteins. We recently showed that expression of misfolded proteins lowers cellular free ubiquitin levels, which compromises the protein degradation machinery. Therefore, the efficient degradation of misfolded proteins is critical to preserve cell health. Cells employ two major mechanisms to degrade misfolded proteins. The first is the ubiquitin-proteasome system (UPS), which ubiquitinates and degrades misfolded proteins with the assistance of segregase Cdc48/p97. The UPS pathway is mainly responsible for the clearance of misfolded proteins present as monomers or smaller aggregates. The second pathway is macroautophagy/autophagy, in which protein aggregates or inclusion bodies are recruited into an autophagosome before transport to the vacuole/lysosome for degradation. This review is focused on the current understanding of the cytotoxicity of misfolded proteins as well as their clearance pathways, with a particular emphasis on mutant Huntingtin.

## 1. Protein Misfolding and Diseases

Misfolded protein aggregates take a variety of forms, with the main types being amorphous aggregates and organized amyloid structures, which are believed to be the cause of many neurodegenerative diseases. Moreover, most of these diseases are attributed to genetic variants in different genes, causing protein misfolding [1]. These misfolded protein aggregates can overwhelm cell quality control and disposal systems. They also cause cytotoxicity by disrupting various cellular membranes or sequestering proteins and mRNA [2,3]. Therefore, a better understanding of the cytotoxicity and clearance of misfolded proteins may provide novel strategies for the prevention and treatment of neurodegenerative diseases.

Huntington’s disease is a genetically inherited neurodegenerative disorder and is caused by a glutamine expansion within the first exon of Huntingtin protein, Htt. The Htt protein in a healthy person has less than 36 glutamine repeats within exon 1, and reduced penetrance is observed when the number of glutamine repeats is between 36 and 39. However, in those with Huntington’s disease, the number of glutamine repeats is more than 39 [4]. Human cells expressing mutant Htt with polyglutamine (polyQ) expansion produce pathogenic exon 1 fragments that contain the polyQ expansion, which is likely a result of proteolysis or alternative splicing [5,6]. This polyQ expansion underlies the protein’s propensity for misfolding and aggregation. Aggregated mutant Htt proteins tend to form an inclusion body, which is a phase-separated and insoluble structure [7]. The resulting aggregates are believed to be toxic to the cell and cause the death of neurons through different mechanisms, but the details are not fully understood [4].

Other important misfolded protein diseases include Alzheimer’s, Parkinson’s, and amyotrophic lateral sclerosis (ALS). The proteins implicated in those diseases (Aβ, α-synuclein, and FUS/SOD1/TDP-43, respectively) are also prone to aggregation, and in the process, cause neurodegenerative or wasting disorders. Alzheimer’s is the most common cause of dementia among older people. In Alzheimer’s, cleavage of amyloid precursor protein generates Aβ, which aggregates to form extracellular plaques and intracellular neurofibrillary tangles. This results in alteration of plasma membrane and synaptic function [8]. Lewy bodies formed from α-synuclein are the causal agent behind Parkinson’s, and its aggregation is associated with a hydrophobic central domain known as non-amyloid β-component domain [9]. FUS/SOD1/TDP-43 work together to form cytoplasmic aggregates in cells of patients afflicted with ALS, but TDP-43 is the most relevant pathological hallmark. The C-terminal domain of TDP-43 was found to be crucial for aggregation [10]. The mechanisms of these diseases and their treatments are a major frontier in biomedical research. In this review, we discuss the nature of protein misfolding, the toxicity of misfolded protein aggregates, and mechanisms by which misfolded protein aggregates are cleared.

## 2. The Nature of Protein Folding and Misfolding

The native (functionally active) conformation of a protein usually has the lowest amount of free energy and is the most thermodynamically stable state [11]. This state is achieved through a series of random fluctuations of the polypeptide chain by Brownian motion, which allows amino acid residues within the polypeptide chain to interact with each other [12,13]. The polypeptide chain is then able to find the lowest energy structure through trial and error and moves down through a funnel-shaped energy landscape until it reaches the lowest energy structure [13]. Furthermore, it is thought that the hydrophobic and polar amino acid residues help determine the general topology of most proteins by “burying” hydrophobic residues in a shell of polar residues. The burying of hydrophobic residues is called “hydrophobic collapse”, which is energetically favorable as it minimizes the surface area of hydrophobic residues and maximizes the surface area of hydrophilic residues [14]. For the folding process of a protein, it first forms a folding nucleus, around which the rest of the structure rapidly condenses as the amount of possible random interactions becomes more and more limited [15]. As the protein structure continues to condense, folds in the polypeptide chain are formed by disulfide bonds between cysteine residues. Higher secondary α-helical and β-sheet structures are stabilized by hydrogen bonds between the amide and carbonyl groups of the main chain on the amino acid, further contributing to the native state conformation and protein stability [16,17].

Chaperones are critical for proteins to achieve their native state conformation, but they are absent in the final structure. Chaperones bind and stabilize unstable proteins with non-native conformations to facilitate correct protein folding, oligomeric assembly, transport, or disposal [18]. Chaperones prevent protein misfolding by avoiding incorrect interactions within and between polypeptides, which increases the yield of folding reactions without any effect on the rate [19]. Cycles of ATP-dependent binding and release of chaperones are required for their functions in protein folding [20]. Chaperones, especially heat-shock proteins (Hsp) in the Hsp70 system, are also responsible for refolding misfolded proteins. This function requires their binding to non-native proteins through a substrate binding domain (SBD), which is mediated by hydrogen bonds and van der Waals contacts with hydrophobic residues [20]. This binding is ATP-dependent and can repeat many times. Proteins that are unable to fold correctly after dissociation from Hsp70 might be targeted to degradation machinery.

Unfolded, partially folded, or misfolded proteins exhibit varying amounts of native-like structure but fail to form a stable hydrophobic core. As a result, these proteins tend to aggregate with other unfolded, partially folded, or misfolded proteins in order to sequester the hydrophobic core in an aqueous environment [18]. This effect is likely exacerbated by the crowded macromolecular environment inside the cell [21]. Nascent proteins may misfold and aggregate during translation. During translation, a peptide must exit the ribosome through a channel that is 100 Å long and 15 Å wide, which means that an average protein domain (50–300 residues) is prohibited from completely folding until the entire peptide has exited the ribosome [22]. As a consequence, this partially folded peptide is exposed to the surrounding aqueous environment and prone to aggregation by attempting to bury the exposed hydrophobic residues with other non-native proteins [23]. Nascent protein aggregation is exacerbated by nearby nascent peptides in polyribosome complexes [24]. Large, multidomain proteins tend to be trapped in intermediate structures, which allows for the stabilization of non-native conformations that expose hydrophobic regions of the peptide chain. These hydrophobic regions sequester themselves away from the aqueous surroundings, and are thus prone to aggregation with each other [23]. However, the presence of chaperones normally prevents this aggregation in vivo [25].

Much more β-sheet structure is detected in aggregated proteins compared to those in their native conformation [1]. These β structures typically amount to 20–25% of the total secondary structure in misfolded protein aggregates, indicating that misfolded protein aggregation likely involves β-sheet-like interactions [26]. Previous studies show that the increase in the amount of β-sheet structure typically comes at the expense of α-helical structure [27]. These β-sheet structures confer a high degree of stability to aggregates because they allow for the formation of a large amount of intermolecular hydrogen bonds, which are energetically favorable. Amyloid fibrils are formed through this mechanism, and grow in a crystal-like fashion as more misfolded proteins are incorporated into the structure through β-sheet hydrogen bonding [28]. The function of the domains flanking the polyQ tract in mutant Htt in amyloid fibril formation has been analyzed in vitro. The N-terminal N17 promotes amyloid fibril formation, while the C-terminal proline rich domain destabilizes fibrils and enhances oligomer formation [29,30]. Amyloid fibrils are important components of inclusion bodies as well. For example, it has been recently elucidated that mutant Htt inclusion bodies are composed of amyloid fibrils [7]. Some misfolded proteins, especially prions, can propagate errant β-sheet structure through a four-rung β-solenoid architecture, which is a protein fold composed of repeating antiparallel β-strands that ultimately form a superhelix. The unpaired uppermost and lowermost rungs can propagate their hydrogen-bonding pattern onto nascent, unfolded proteins, causing nascent proteins to misfold [31].

## 3. Cytotoxicity of Misfolded Protein Aggregates

### 3.1. Membrane Damage by Misfolded Protein Aggregates

Misfolded protein aggregates, such as Aβ, have been shown to interact with cell membranes and synthetic phospholipid bilayers, which results in the destabilization of the membrane or dysfunction of membrane-bound proteins in vivo and in vitro [32,33,34]. The interaction of Aβ with membranes could induce the formation of non-specific, “leaky” pores in the plasma membrane [2]. The assembly of misfolded proteins on the cellular membrane induces pore formation that can lead to an increase in intracellular ion concentration, with one particular ion of interest being Ca^2+^ [33]. Higher intracellular Ca^2+^ levels result in upregulation of ion effluxer activity in an attempt to normalize the ion gradient. ATP production is subsequently increased to power upregulated effluxer activity, which also results in an increase in reactive oxygen species (ROS), a byproduct of ATP synthesis [35]. High levels of ROS can further damage lipids, nucleic acids, and proteins to confer cytotoxicity [36]. In support of this notion, it has been shown that cells were protected against the cytotoxicity of misfolded protein aggregates when treated with antioxidants [37].

Misfolded protein aggregates also interact with different organelles and cause detrimental effects. The mutant Htt inclusion body is of particular interest because its structure has recently been elucidated via cryo-electron tomography [7]. The mutant Htt inclusion body consists of an amyloid core with flanking regions that protrude in a bottlebrush fashion. It is these protrusions that interact with cellular membranes and impair their functions. One particularly affected organelle is the endoplasmic reticulum (ER) because the mutant Htt fibril protrusions are able to drive lipid-mediated reorganization of the ER network at their periphery. This reorganization was characterized by extremely high amounts of membrane curvature present at sites of inclusion body interface with the ER [7]. This interface is supported by previous in vitro studies showing that mutant Htt interacts with synthetic vesicles and lipid membranes via the 17 residues at the N-terminus of Htt exon 1 [30,38,39]. Moreover, this study found that regions of the ER surrounding the inclusion body showed reduced membrane dynamics, which could affect downstream cellular processes that depend on a highly dynamic ER. Protein translation is also likely halted in ER regions affected by mutant Htt fibrils, since the portions of the ER in contact with the fibrils lack ribosomes and Sec61 translocons. Finally, ER-associated protein degradation (ERAD) factors and chaperones accumulate in regions affected by the fibrils, indicating that mutant Htt inclusion bodies can cause accumulation of misfolded proteins at the ER [7]. This is consistent with a previous observation that mutant Htt expression results in ERAD dysfunction [40].

### 3.2. Accumulation of Misfolded Protein Aggregates Alters Ubiquitin Homeostasis

Prior to proteasomal degradation, misfolded proteins undergo ubiquitination via ubiquitin enzymes [41]. Once the designated proteins are transported to the proteasome, the conjugated ubiquitin is recycled and rejoins the free ubiquitin pool, which depends on substrate deubiquitination by proteasome-associated metalloprotease Rpn11 and cysteine proteases Ubp6 and Uch2/Uch37 [42,43]. However, the accumulation of aggregated misfolded proteins reduces ubiquitin recycling and leads to cytotoxicity by impairing UPS function. Cdc48 is a segregase/unfoldase and is responsible for the disaggregation and unfolding of ubiquitinated misfolded protein aggregates. The ubiquitin hydrolases Doa4 and Otu1 associate with Cdc48 and are possibly responsible for ubiquitin recycling from ubiquitinated proteins [44,45,46]. We found that cells with dysfunctional Cdc48 exhibited an increase of ubiquitinated proteins and protein aggregation, but showed decreased free ubiquitin levels, which leads to impaired degradation of other proteins via the UPS [47]. Interestingly, deletion of San1 and Ubr1, two E3 ubiquitin ligases responsible for the ubiquitination of misfolded proteins, partially suppressed the growth defect in cells with dysfunctional Cdc48. The slow growth phenotype of *cdc48* mutant cells was also rescued by the overexpression of ubiquitin [47]. These results support the model that misfolded protein aggregation soaks up free ubiquitin like a sponge, leading to cytotoxicity by slowing ubiquitin-dependent proteasomal degradation of other proteins. Previous studies showed that dysfunctional Cdc48 results in the accumulation of ubiquitinated substrates on proteasomes, but it is still uncertain if this accumulation impairs UPS function directly [47,48,49].

### 3.3. Sequestration of Functional Proteins by Misfolded Protein Aggregates

Aggregation of normal functional proteins, along with misfolded proteins, also contributes to cytotoxicity, with stress granules and P-bodies constituting pertinent examples. Stress granules and P-bodies in yeast are membrane-less aggregates formed during cell stress, consisting of mRNA molecules and proteins [50]. Their formation in yeast is promoted by the dimerization of Edc3, an RNA-binding protein. Moreover, Lsm4p, a protein involved in mRNA decay, contains an intrinsically disordered region which also contributes to stress granule and P-body formation [50]. It is speculated that mRNAs act as a scaffold for RNA-binding proteins to form stress granules. Thus, stress granules are the location to store translationally silent mRNA, acting as a transfer site for mRNAs destined for degradation or targeted to polysomes for translation [51]. The sequestration of RNA-binding proteins will limit their role in other cellular processes, causing cytotoxicity [3,52]. In addition, stress granule formation likely contributes to the development of various neurodegenerative disorders by harboring misfolded proteins and serving as precursors to more pathological protein aggregates [51,53]. Therefore, stress granule formation may confer cytotoxicity by sequestering functional proteins and mRNA and facilitating insoluble aggregate formation.

### 3.4. Misfolded Protein Aggregates Alter Organelle Homeostasis

Organelle homeostasis can be affected by misfolded protein aggregates. Increased Ca^2+^ levels driven by aggregate-induced pore formation on the cellular membrane contributes to dysregulation of homeostasis through the accumulation of Ca^2+^ in mitochondria. This accumulation causes the mitochondria to swell and induces pore formation, which releases apoptotic factors such as cytochrome c, further contributing to cytotoxicity [54]. Lysosomal homeostasis can be impaired in Alzheimer’s disease, in which Aβ accumulates in the lysosome and alters its permeability. This in turn results in the release of Aβ into the cytosol [55]. In addition, Parkinson’s disease-associated mutations in several genes have been shown to negatively affect lysosomal activity [56]. As mentioned above, some misfolded protein aggregates interact with ER membrane, causing ER stress and altered ER homeostasis. In addition, α-synuclein induces ER stress, impairs ER-Golgi transport function, and causes Golgi fragmentation [57].

## 4. Clearance of Misfolded Proteins

### 4.1. Misfolded Protein Degradation by the Ubiquitin-Proteasome System (UPS)

Cells need to maintain protein homeostasis for their growth and survival, and one way to achieve this goal is through protein degradation. The 26S proteasome is the foremost protein degradation complex in eukaryotic cells, and functions in both cytosol and nucleus [58]. The proteasome consists of a barrel-shaped 20S core particle in the center and two 19S regulatory particles at both ends. In growing cells, the proteasome is responsible for at least 80% of protein turnover [59]. Proteins are targeted to the 26S proteasome for degradation only after their ubiquitination, a process that attaches a ubiquitin chain to the target protein. In the process of protein ubiquitination, the ubiquitin-activating enzyme (E1) catalyzes ATP-dependent formation of a thioester bond with the main-chain carboxyl group of the terminal glycine residue of ubiquitin [41]. The activated ubiquitin is then transferred to a ubiquitin-conjugating enzyme (E2) via transesterification. In the last step, ubiquitin ligases (E3) attach ubiquitin by the formation of isopeptide bonds between ε-amino groups of lysine residues in target proteins and the activated carboxyl group of ubiquitin [41]. This process can repeat through sequential rounds of conjugation, which results in polyubiquitination of target proteins. The wide array of E3 enzymes (~80 in yeast) confer selectivity of their ubiquitination targets by directly binding to specific substrates. This stands in contrast with the relatively fewer E1 (one in yeast) and E2 (11 in yeast) enzymes [41]. The E3 ligases specific for misfolded proteins are San1 and Ubr1 in budding yeast. Misfolded proteins in the nucleus are ubiquitinated by San1 [60], and its counterpart Ubr1 mainly functions in the cytosol [61]. San1 binds directly to misfolded protein substrates, but Ubr1 appears to rely on chaperones for its binding to misfolded substrates [62,63].

After ubiquitination, misfolded protein aggregates undergo disaggregation or unfolding into monomers to enable their degradation by the proteasome. This disaggregation depends on the conserved AAA ATPase Cdc48 (p97/VCP in higher eukaryotes) [41]. Six Cdc48 monomers form a double-ring structure surrounding a central pore. Each Cdc48 monomer consists of an N-terminal domain, a C-terminal tail, and two centrally located ATPase domains [64]. The Cdc48 complex also includes two essential cofactors, Ufd1 and Npl4, which contain ubiquitin-binding domains for the recruitment of ubiquitinated proteins [64]. The Cdc48 complex segregates/unfolds misfolded proteins by passing its clients through the central pore. In addition, the Cdc48/p97 complex has been shown to regulate autophagy by stabilizing Beclin-1 through stimulation of ataxin-3. Ataxin-3 deubiquitinates and stabilizes Beclin-1 in a Cdc48/p97-dependent manner. This allows Beclin-1 to be incorporated into phosphatidylinositol-3-kinase (PI3K) complex I, which in turn enhances production of phosphatidylinositol-3-phosphate (PI(3)P), a key signaling lipid that recruits downstream autophagy machinery such as Atg8/LC3 [65]. Cdc48/p97 also plays a role in lysosome degradation, in which it modulates ubiquitin chains on damaged lysosomes, thus targeting them for degradation [65].

After release from the Cdc48 complex, the ubiquitinated protein first interacts with the proteasomal 19S regulatory particle through polyubiquitin chains [66]. The 19S regulatory particle contains ubiquitin-interacting motifs and pleckstrin homology domains that facilitate ubiquitin binding. In addition, the 19S regulatory particle also recognizes the shuttling receptors, and these receptors in budding yeast include Rad23, Dsk2, and Ddi1. The ubiquitin-like domains on these receptors bind to the proteasome, and their ubiquitin-associated domains bind to ubiquitin, thereby serving as a mediator between substrate proteins and the proteasome [67]. After proteasome binding, structural changes in the 19S particle facilitate the movement of the substrate protein into the central tunnel of the 20S complex, which is followed by procedural degradation of substrate proteins by cleavage protein into small peptides [66]. These small peptides are degraded further into amino acids by cytosolic endopeptidases and aminopeptidases [68].

Expression of mutated Htt with a polyglutamine expansion may impair UPS function, since the 26S proteasome has difficulty digesting these glutamine sequences [69]. On the other hand, proteasome inhibitors increase the level of mutant Htt, accelerating the formation of inclusion bodies [70]. Results using mouse models have shown that the clearance of mutant Htt aggregates was blocked when the cell culture was treated with proteasome inhibitors [71]. Proteasomes can also become sequestered into inclusion bodies, but these proteasomes remain functionally active, supporting the notion that this sequestration does not impair proteasome function [72,73].

To summarize, misfolded protein aggregates can be degraded by the UPS to limit their cytotoxicity and preserve protein homeostasis. This degradative pathway begins with the polyubiquitination of misfolded proteins by San1 and Ubr1 in budding yeast. Ubiquitinated misfolded protein substrates are then recruited to the Cdc48 complex for disaggregation/unfolding, which prepares them for targeting to the proteasome. At the proteasome, misfolded protein substrates are recognized and deubiquitinated by the 19S regulatory particles and degraded by the central 20S proteasome core particle. Substrate proteins are released from the proteasome as small polypeptides and are degraded further into individual amino acid residues in the cytoplasm (Figure 1).

### 4.2. Autophagy

Macroautophagy (further referred to as autophagy) is a highly conserved cellular process that degrades and recycles cellular structures and proteins. This process is critical for the maintenance of cell homeostasis under starvation and vegetative conditions [74]. Three types of autophagy are starvation-induced non-selective autophagy, selective autophagy, and chaperone-mediated autophagy (CMA). Starvation-induced autophagy is triggered when cells encounter nitrogen-deficient (starvation) conditions. Cells respond by engulfing a portion of their cytosolic contents into a structure called the autophagosome. The autophagosome is a double-membraned vesicle that sequesters autophagic cargoes and delivers them to the vacuole (lysosome in mammals) for degradation [74]. Selective autophagy, on the other hand, can be induced in both vegetative and starvation conditions and involves the engulfment of specific cargoes into the autophagosome, which is then delivered to the vacuole for degradation. Cargoes targeted by selective autophagy include misfolded protein aggregates and various organelles. Selective autophagy receptors (SARs) mediate the interaction between the specific cargo and the core autophagy machinery [75,76]. The focus of this review will be the clearance of misfolded proteins by the autophagy pathways.

#### 4.2.1. Non-Selective and Selective Autophagy

Starvation-induced non-selective autophagy typically begins with the inactivation of TORC1 kinase by limited nutrient availability. Activated TORC1 phosphorylates the core autophagy protein Atg13, preventing its association with other autophagy proteins, namely Atg1 and Atg17 [77]. When TORC1 becomes inactive, dephosphorylated Atg13 interacts with Atg1 and Atg17 [78]. This interaction marks the formation of the phagophore assembly site (PAS), which is partly organized by scaffold proteins Atg11 and Atg17 [79]. The PAS further recruits Atg9 vesicles to initiate the nucleation and expansion of the double-membrane phagophore, which eventually matures into an autophagosome [80]. Atg8-phosphatidylethanolamine (Atg8-PE) is a major factor in autophagosome membrane expansion and is required in all types of autophagy [80]. The autophagy proteins work together to expand the membrane and engulf the cargo, resulting in autophagosome formation. For non-selective autophagy, the cargo is non-specific and typically contains varied cytoplasmic contents. After autophagosome-vacuole fusion, the autophagosomal membrane is degraded by the lipase Atg15 and the contents are destroyed and recycled by vacuolar hydrolases [81].

Selective autophagy can occur under either starvation or vegetative conditions [82]. SARs are responsible for the recognition of specific cellular components by the autophagy machinery [83], and this process requires the activation of an SAR via phosphorylation by a casein kinase [80]. SARs are usually found on the membrane of organelles, with exceptions for Cue5 and Ubx5, which are discussed below. In budding yeast, after phosphorylation, SARs bind to the scaffold protein Atg11 through an Atg11-binding region [80]. This binding in turn recruits the core autophagy proteins to drive phagophore expansion [80]. The phagophore then envelops the targeted cargo to form an autophagosome that fuses with the vacuole for degradation of the enclosed cargo [84]. Important yeast SARs include Atg32 (mitophagy), Atg36 (pexophagy), Atg39 (nucleophagy), and Atg40 (ER-phagy). Selective autophagy in mammalian cells utilizes functional counterparts of these SARs, including p62/SQSTM1, which is critical for the autophagy of ubiquitinated substrates [80]. Other important mammalian SARs include OPTN/NDP52 (mitophagy), NBR1 (pexophagy), and CCPG1/FAM134B (ER-phagy) [83].

#### 4.2.2. Cue5/Tollip-Mediated Autophagy of Misfolded Protein Aggregates

Aggrephagy is the selective autophagic degradation of misfolded protein aggregates. Misfolded proteins are prone to aggregation, and their disposal is critical to cell health. The SAR for aggrephagy in budding yeast is Cue5, which contains a ubiquitin-binding CUE domain and an Atg8-interacting motif (AIM). It has been shown that Cue5-mediated aggrephagy of mutated Htt proteins requires their ubiquitination by E3 ligase Rsp5 [63,85]. Cue5 binds to these ubiquitinated aggregates through its CUE domain, then recruits these aggregates to phagophore-localized Atg8 through its AIM, which enables the engulfment of protein aggregates by the phagophore [85,86]. After engulfment, the mature autophagosome is targeted to the vacuole for degradation. It should be noted that Cue5 is the only known yeast SAR that does not interact with scaffold protein Atg11 during autophagy initiation [80], thus it remains unclear how Cue5 induces aggrephagy without the assistance of Atg11. The mammalian homolog of Cue5 is TOLLIP, which also contains a ubiquitin-binding CUE domain and an LC3-interacting region (LIR) [85,87]. LC3 is the mammalian homolog of Atg8, and TOLLIP directly targets aggregated proteins to LC3-coated phagophores. In addition to TOLLIP, several other aggrephagy SARs have been identified in mammalian cells, including p62/SQSTM1, NBR1, and OPTN [88]. These SARs also have ubiquitin-binding domains and LIRs to link the ubiquitinated cargo to LC3-coated phagophores [88]. Currently, it is unclear why many different SARs are involved in aggrephagy, and one possibility is that these SARs show substrate specificity (Figure 1).

#### 4.2.3. Ubx5-Dependent Autophagy for Damaged Cdc48 Complexes

A recent study demonstrated an autophagic pathway for damaged Cdc48 complexes, and a Cdc48 cofactor, Ubx5, acts as the SAR for this pathway. Ubx5 contains a ubiquitin-interacting motif (UIM) that binds directly to the UIM-docking site (UDS) on Atg8 for autophagic degradation of damaged Cdc48 [89]. Strikingly, the UDS in Atg8 is different from the AIM for other SARs, indicating the presence of two docking sites in Atg8 for SARs. Ubx5-dependent autophagy could be induced by either nitrogen starvation or Cdc48 inactivation. Ubx5 binds to Cdc48 and targets the Cdc48 cargo to the phagophore by binding to Atg8 through its UIM. The Cdc48-containing autophagosome is then targeted to the vacuole for degradation [89]. As the Cdc48 complex associates with misfolded protein aggregates for their disaggregation [47], it would be interesting to test if misfolded proteins are also targeted to the phagophore along with Cdc48 for autophagic clearance.

#### 4.2.4. The Role of Chaperone Complex HspB8-Bag3 in Autophagic Clearance of Misfolded Proteins

Chaperones have the ability to target misfolded proteins for degradation, including chaperone-mediated autophagy (CMA). The HspB8 and Bag3 chaperone complex is involved in chaperone-assisted selective autophagy of misfolded proteins. Mutations in HspB8 chaperone are associated with neurodegenerative disorders. Bag3 is a co-chaperone responsible for connecting chaperones to different protein clearance pathways and stimulating autophagy when proteasome activity is insufficient [90]. In the HspB8-Bag3 chaperone complex, HspB8 recognizes ubiquitinated misfolded proteins and their aggregates, and Bag3 stimulates autophagy [91]. Bag3 facilitates the interaction between the cargo-associated HspB8 and p62/SQSTM1, a mammalian SAR. In addition, the HspB8-Bag3 complex helps sequestrate misfolded proteins, which enhances their autophagic clearance and reduces their toxicity [90]. Bag3 has also been shown to work with 14-3-3 for sequestration of misfolded proteins [92], but it is still unknown whether HspB8 is involved in this process.

#### 4.2.5. Inclusion Body Autophagy

Mutant Htt proteins can form soluble aggregates or insoluble phase-separated inclusion bodies. Autophagic clearance of mutant Htt inclusion bodies is supported by the observation that autophagy-related proteins, such as Atg8/LC3, Atg5, Atg12, and Atg16, are recruited to mutant Htt inclusion bodies in mammalian cells. Moreover, the level of mutant Htt aggregation increased when Atg8/LC3 or Atg5 were knocked down with siRNAs [93]. In addition, conditional knockout of Atg5 or Atg7 in mouse neurons causes neurodegeneration and premature formation of ubiquitin-positive inclusion bodies [94]. One study shows the colocalization of p62 with mutant Htt aggregates in *Drosophila* [95]. However, further studies are required to clarify whether the observed autophagy is for misfolded protein aggregates or inclusion bodies (Figure 1).

Expression of mutated Htt with polyglutamine repeats in yeast cells causes inclusion body formation [7,96]. Using budding yeast as a model system, we demonstrated inclusion body autophagy by showing vacuolar localization of Htt103QP-GFP, a mutated Htt with a 103 polyglutamine expansion, as well as the disappearance of cytoplasmic inclusion bodies. Moreover, this process depends on Atg1 and Atg8 [97,98]. Interestingly, the SAR Cue5 was shown to be required for the autophagy of mutant Htt in yeast cells [85], but it remains to be determined if the Cue5-dependent autophagy is specific for mutant Htt aggregates or inclusion bodies. Thus, the identity of SARs specific for inclusion body autophagy remains unclear.

Dsk2 is a yeast proteasome shuttling receptor, and we found impaired inclusion body formation in yeast cells lacking Dsk2, which decreases the clearance efficiency for mutated Htt through inclusion body autophagy. This result indicates that Dsk2 facilitates the autophagic degradation of mutant Htt by enhancing inclusion body formation [97]. Dsk2 is the yeast homologue of human ubiquilin 2 (UBQLN2), and mice expressing mutant UBQLN2 show increased insoluble and ubiquitin-positive inclusion bodies that colocalize with p62 [99]. Mutations in the *UBQLN2* gene are directly linked to the development of ALS [100], and inactivation of UBQLN2 expression in HeLa cells reduced autophagic flux and autophagosome acidification [101]. However, it remains unclear if UBQLN2 also promotes inclusion body formation to facilitate the autophagy of misfolded proteins in mammalian cells. Impaired inclusion body formation and autophagy was also seen in cells lacking heat-shock proteins, such as Hsp70 [98]. In addition, p62 and 14-3-3, a family of conserved regulatory proteins, also facilitate inclusion body formation, but their role in inclusion body autophagy is yet to be determined [92,102].

Stress granules and P-bodies are membrane-less structures formed through interactions between mRNAs and mRNA-binding proteins, and these structures facilitate the formation of inclusion bodies [50]. A previous study found that stress granules and P-bodies are cleared via autophagy in a process called granulophagy. This notion is supported by the vacuolar localization of GFP-tagged components from stress granules and P-bodies in both yeast and mammalian cells. Interestingly, functional Cdc48 is required for granulophagy [103]. A previous study shows that misfolded proteins, such as ALS-linked variants of SOD1, specifically accumulate and aggregate within human stress granules [53]. One open question is whether stress granules and P-bodies also play a role in the autophagic clearance of misfolded proteins. Moreover, it is unclear if granulophagy and inclusion body autophagy share a common pathway.

## 5. Conclusions

Protein homeostasis is vital to the health of the cell, and many different mechanisms are dedicated to the maintenance of protein homeostasis. However, threats to this homeostasis arise when misfolded proteins are produced. Misfolded proteins are characterized by their inability to form a stable hydrophobic core, leading to aggregation with other misfolded proteins. The main degradation pathways for misfolded proteins and their aggregates include the UPS and autophagy. Prior to UPS-mediated degradation, the Cdc48 complex is required for disaggregating/unfolding misfolded protein aggregates. For larger insoluble aggregates and inclusion bodies, autophagy plays a more important role in their clearance, but an important open question is how SARs enable their recognition by the autophagy machinery.

As cells age, protein homeostasis becomes less efficient, and the subsequent accumulation of misfolded proteins and their aggregates causes disease, such as neurodegenerative disorders, through a variety of cytotoxic effects. The accumulation of misfolded proteins confers cytotoxicity by overwhelming protein disposal systems, damaging cellular membranes that subsequently increases ROS production, and sequestering normal functional proteins. Strategies to limit the cytotoxicity of misfolded proteins include upregulating their clearance and minimizing their cytotoxic effects. Further effort is required to translate scientific findings into medical treatments.

## Figures and Tables

**Figure 1 cells-10-02835-f001:**
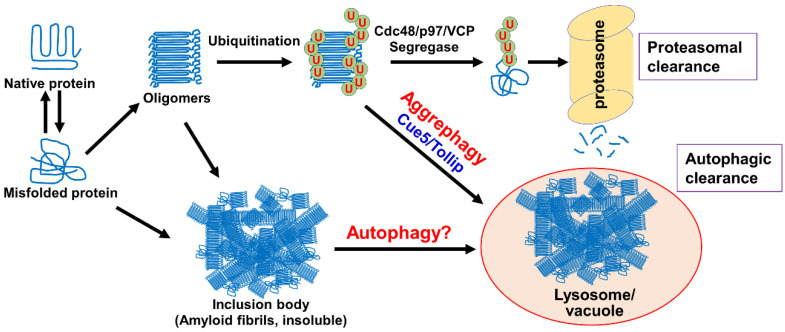
The clearance of misfolded proteins by the ubiquitin proteasome system and autophagy.

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
