# Peer review of "The Cytotoxicity and Clearance of Mutant Huntingtin and Other Misfolded Proteins"

_cells, 2021, doi:10.3390/cells10112835_

Round 1
Reviewer 1 Report
In this review manuscript by Folger and Wang, the toxicity and clearance mechanism of protein aggregates are discussed. The review manuscript is well organized and nicely readable. Only minor points are suggested for the author's consideration.
1, I suggest to add a little more description on diseases caused by protein aggregates other than Htt. In other words, to expand the contents between line 47-53.
2, I suggest add a small section to describe the current treatments, pre-clinical or clinical trials for disease such as Huntington Disease.
3, The authors may consider add a figure to show the different pathways for clearance of protein aggregates.
Author Response
In this review manuscript by Folger and Wang, the toxicity and clearance mechanism of protein aggregates are discussed. The review manuscript is well organized and nicely readable. Only minor points are suggested for the author's consideration.
1, I suggest to add a little more description on diseases caused by protein aggregates other than Htt. In other words, to expand the contents between line 47-53.
►Yes, we added a brief description of these diseases (line 51-57 in the revised version).
2, I suggest add a small section to describe the current treatments, pre-clinical or clinical trials for disease such as Huntington Disease.
►We added a new section to describe the treatment (line 188-198 in the revised version).
3, The authors may consider add a figure to show the different pathways for clearance of protein aggregates.
►Good point. This is also the suggestion from the editor. We created a graphic model to illustrate the clearance of misfolded proteins by the ubiquitin-proteasome system and autophagy (line 255-256).
Reviewer 2 Report
In the present review, the authors described the citotoxicity and clearance of misfolded protein aggregates. The review is well written and structured.
I have some comments about this review:
1) the authors described only the mHTT as example for misfolded protein aggregates. It would be useful to cite other examples of misfolded proteins toxicity and clearance(e.g. other polyQ proteins, proteins related to ALS, PD or other NDs).
2) protein aggregates might cause citotoxicity in different ways, some are here described. It would be useful to described also alterations in the organelle homeostasis (e.g. mithocondria, lysosome)
3) CASA complex (BAG3 and HSPB8) might be involved in the autophagic clearance of misfolded protein aggregates, please described in the text this part.
4) the authors cite cdc48/p97 as protein involved in disaggregation. Recently this protein was found to be involved in autophagic pathway as regulator of Beclin-1 and in the organelle specific degradation of damaged lysosome. I suggest to add this part in the review.
Author Response
1) the authors described only the mHTT as example for misfolded protein aggregates. It would be useful to cite other examples of misfolded proteins toxicity and clearance (e.g. other polyQ proteins, proteins related to ALS, PD or other NDs).
►The editor also noticed the clear focus on huntingtin and way-less words for other amyloidogenic proteins in this manuscript. The suggestion from the editor is to formulate a more suitable title. As suggested, we have added “mutant Huntingtin” to the title so that it reflects more accurately the content of this review. Accordingly, we have made minor adjustments to the wording in the abstract.
2) protein aggregates might cause cytotoxicity in different ways, some are here described. It would be useful to described also alterations in the organelle homeostasis (e.g. mitochondria, lysosome)
►As suggested, we added a new section to discuss the effect of misfolded proteins on organelle homeostasis (line 177-187).
3) CASA complex (BAG3 and HSPB8) might be involved in the autophagic clearance of misfolded protein aggregates, please described in the text this part.
►A new section was added for the function of Hspb8-Bag3 complex in autophagy (line 320-330).
4) the authors cite cdc48/p97 as protein involved in disaggregation. Recently this protein was found to be involved in autophagic pathway as regulator of Beclin-1 and in the organelle specific degradation of damaged lysosome. I suggest to add this part in the review.
►This part was added in the revised version (line 224-230).
Round 2
Reviewer 2 Report
The Authors have answered to all the suggestion required by the Reviewer.
Author Response
Thanks!